# Attention-Based Active Learning Framework for Segmentation of Breast Cancer in Mammograms

Xianjun Fu [1], Hao Cao [2], Hexuan Hu [2], Bobo Lian [1], Yansong Wang [2], Qian Huang [2] and Yirui Wu [2,3,*]

[1] School of Artificial Intelligence, Zhejiang College of Security Technology, Wenzhou 325016, China
[2] College of Computer and Information, Hohai University, Nanjing 210093, China
[3] Key Laboratory of Symbolic Computation and Knowledge Engineering of Ministry of Education, Jilin University, Changchun 130015, China
* Correspondence: wuyirui@hhu.edu.cn

**Abstract:** Breast cancer is one of most serious malignant tumors that affect women's health. To carry out the early screening of breast cancer, mammography provides breast cancer images for doctors' efficient diagnosis. However, breast cancer lumps can vary in size and shape, bringing difficulties for the accurate recognition of both humans and machines. Moreover, the annotation of such images requires expert medical knowledge, which increases the cost of collecting datasets to boost the performance of deep learning methods. To alleviate these problems, we propose an attention-based active learning framework for breast cancer segmentation in mammograms; the framework consists of a basic breast cancer segmentation model, an attention-based sampling scheme and an active learning strategy for labelling. The basic segmentation model performs multi-scale feature fusion and enhancement on the basis of UNet, thus improving the distinguishing representation capability of the extracted features for further segmentation. Afterwards, the proposed attention-based sampling scheme assigns different weights for unlabeled breast cancer images by evaluating their uncertainty with the basic segmentation model. Finally, the active learning strategy selects unlabeled images with the highest weights for manual labeling, thus boosting the performance of the basic segmentation model via retraining with new labeled samples. Testing on four datasets, experimental results show that the proposed framework could greatly improve segmentation accuracy by about 15% compared with an existing method, while largely decreasing the cost of data annotation.

**Keywords:** breast cancer; image segmentation; active learning; deep learning

## 1. Introduction

Breast cancer is one of the most serious malignant tumors that threatens the health of women. It is reported that about 12.5 percent of women are affected by breast cancer worldwide [1]. In China, the incidence of breast cancer is increasing by 0.5% per year [2], which has become one of the most dangerous and deadly diseases for women's health. Early-stage breast cancer screening can aid in detecting disease early, thus largely increasing the chances of recovery [3]. These screenings are conducted by experienced doctors, who check the existence of malignant lesions in mammograms for further diagnosis and evaluation. However, such operations are not only annoying and time-consuming, but also carry the risk of wrong or missed detections with manual examinations [4], which is reported to be as high as 30% in breast cancer screening [5]. Since the advent of artificial intelligence (AI), researchers have adopted computer-aided detection (CAD) technologies to considerably reduce the amount of work involved in breast cancer screening. In the early stages of this application, CAD systems with traditional image processing algorithms were applied to aid in breast cancer diagnosis (masses, microcalcification etc.) [6]. With the rapid development of deep learning, the accuracy of medical image segmentation, regarded as a subfield of image segmentation, has been significantly improved. For example, Wang et al. [7] proposed a breast tumor semantic segmentation method based on a

convolutional neural network (CNN), which provided accurate results with the extracted biomarkers in radiology imaging. Later, Hann et al. [8] utilized a multiple UNet network to generate multiple segmentations for fusion, which were further thresholded to generate the final segmentation result.

However, these impressive achievements of deep learning in breast cancer screening were built on a large amount of labeled data, and accuracy would greatly drop without it [9]. Due to privacy issues, it is difficult to obtain sufficient breast cancer images for labeling. Moreover, labeling generally requires medical experts with professional knowledge and experience, making it time-consuming and costly to acquire enough labeled data. Achieving desirable segmentation results with fewer samples has thus recently become a research focus.

Benefiting from its ability to achieve high performance with few labeled samples, active learning has become a feasible approach for less-thoroughly labeled breast cancer data [10]. Several successful applications have been implemented to show the remarkable power of active learning in medical image analysis. For example, Vishwesh et al. [11] combined an active learning framework with deep learning for medical image segmentation, where new active learning strategies guided the segmentation model to learn the diversity of uncertain and unlabeled data, thus greatly achieving convergence in accuracy with less labelling. Later, Li et al. [12] proposed a novel active learning framework for histopathology image analysis, where two groups of unlabeled data were selected in each training iteration, one annotated by experts and the other selected from high-confidence unlabeled samples to assign pseudo-labels. Both manual labeling and pseudo-label generation Were able to largely alleviate the problem of scarce labeled samples.

Based on the advantages of using deep learning and active learning for automatic screening tasks, we propose an attention-based active breast cancer segmentation model which is capable of achieving desirable segmentation results without a high quantity of labeled images. The proposed model consists of a basic segmentation model, an attention-based sampling scheme and an active learning based labeling strategy. Specifically, a multi-scale fusion and enhancement module based on UNet is first adopted for segmentation. Afterwards, a novel attention mechanism is used to evaluate the similarity between the unlabeled and segmented samples, thus offering weights as criteria to measure the uncertainty or informativeness of unlabeled samples with respect to the trained and basic segmentation model. Finally, an active learning strategy is used to sort unlabeled samples with weights, thus determining selections that need to be further manually labelled. We observed that these selected samples often contained appearance or shape features which were unacknowledged by the basic segmentation model subjected to the current training dataset. With iterations of retraining with the most informative unlabeled samples, the proposed model stably approached the upper bound of segmentation performance for high accuracy.

The contributions of this paper are summarized as follows:

- We propose an attention-based active breast cancer segmentation framework which effectively improved the accuracy of segmentation with few training samples, thus alleviating the high cost of labeling breast cancer images.
- A novel attention-based sampling scheme is proposed which measures the most informative unlabeled samples via calculating similarity weights.
- We adopt an active learning strategy for the global optimization of accuracy performance, which iteratively selects appropriate unlabeled samples to first manually label and then retrain the model to boost performance.

The rest of this paper is organized as follows. Section 2 reviews related work on the segmentation of breast cancer images. Section 3 presents an overview of the proposed method. Details of the basic segmentation model, attention-based sampling scheme and active-learning-based labeling strategy are also discussed in Section 3. Section 4 presents and discusses the experimental results. Finally, Section 5 concludes the paper.

## 2. Related Work

In this section, we give a brief literature review, including prior works on traditional segmentation methods for breast cancer images, deep-learning-based segmentation methods for breast cancer images, as well as active learning methods.

### 2.1. Traditional Segmentation Methods for Breast Cancer Images

Traditional segmentation methods usually apply image processing technologies for segmentation. However, they generally suffer from the drawbacks of low accuracy and are sensitive to the quality of sampled images.

For example, Cheng et al. [13] proposed a near-automatic ultrasound image segmentation algorithm which builds a solid foundation of computer-aided diagnosis for breast cancer. Later, Eziddin et al. [14] proposed the segmentation of mammograms using an iterative fusion process of information obtained from multiple knowledge sources, including context information, image processing algorithms, prior knowledge and so on. Gnonnou et al. [15] proposed a structural method to separate breast margins at pixel-level, thus accurately extracting tumor regions. Later, Kaushal et al. [16] proposed an automated segmentation technique followed by self-driven post-processing operations to detect cancerous cells effectively. Recently, Jing et al. [17] proposed a simple but effective segmentation method with the concept of global thresholding, which successfully segmented tumor regions in breast histopathology images. During the process, partial contrast stretching and median filtering are specially designed to improve image quality for segmentation.

### 2.2. Deep-Learning-Based Segmentation Methods for Breast Cancer Images

Inspired by the remarkable performance of deep learning methods in image classification and segmentation tasks [18,19], researchers have proposed several works on the segmentation of breast cancer images with various kinds of networks.

For example, Su et al. [20] proposed a fast scanning deep convolutional neural network (FCNN) to achieve pixel-wise region segmentation, successfully eliminating the redundant computation of the original CNN without sacrificing performance. Later, Simin et al. [21] proposed the combination of deep learning with traditional features for medical image classification, where a CNN model is first used to extract image features, and then support vector machines are used for feature learning and classification. Then, Roy et al. [22] used a dropout strategy to generate different Monte Carlo segmentations, where they computed the dissimilarity of these segmentations to measure the structural uncertainty of the image. In such way, they could confidently choose the best matching segmentations from candidates for output.

Despite their strengths, the shortage of sufficient training data affects the performance of deep learning methods, and researchers have focused on active learning to improve the effectiveness of deep learning. For example, Shen et al. [23] proposed a novel deep active learning model for the image segmentation of breast cancer on immunohistochemistry images. They not only achieved significant performance improvements in the segmentation of breast cancer images, but the system also showed promise for implementation as a real-world application.

Recently, difference comparisons between multiple candidate segmentation maps has become an effective method for sampling in active learning. For example, Wang et al. [24] believe that easy samples tend to obtain similar segmentations in K models, where they use K different models to segment images and measure the similarity of outputs, thus building connections between different models for further comparisons. Recently, Zhang et al. [25] generated two segmentation maps before and after processing of their proposed attention module, thus calculating the similarity coefficient of maps to guide sampling of their active learning framework.

### 2.3. Active Learning Methods

To reduce the cost of labeling, active learning selects the most valuable and informative samples from unlabeled samples for the labeling task, which relieves the dependence of deep learning models on large training datasets. Due to its ideal function in achieving desirable performance with few labelled samples, researchers in the medical image analysis community have proposed several methods for CAD.

For example, Ayerdi et al. [26] proposed an interactive image segmentation system using active learning, which allows rapid segmentation without the requirement of manual intervention. Later, Sharma et al. [27] adopted active learning to perform biomedical image segmentation with limited labeled data, where they combined UNet and an active learning query strategy to select additional samples for annotation, thus capturing the most uncertain and representative samples. Then, Li et al. [28] proposed a deep active learning framework which combines an attention-gated fully convolutional network (ag-FCN) and a distributional difference-based active learning algorithm (dd-AL) to iteratively annotate samples. Later, Lai et al. [29] proposed a semi-supervised active learning framework with region-based selection criteria which iteratively selects regions for annotation queries to rapidly expand the diversity and number of marker sets.

Most recently, Gaillochet et al. [30] proposed a test-time augmentation method for active learning in medical image segmentation, which exploits the power of uncertain information provided by data transformation. Bai et al. [31] proposed a difference-based active learning (DEAL) method for bleed segmentation, which successfully bridged the gap between class activation maps (CAMs) and ground truth with few annotations.

## 3. Method

To deal with the high labeling cost of breast cancer images, we propose an attention-based deep active learning framework for segmentation in mammograms. First, the overall structure is given, offering a global view of how the proposed framework works. Then, we present the basic breast cancer segmentation model with multi-scale feature fusion and enhancement. Afterwards, we describe an attention-based sampling scheme to assign weights for unlabeled samples under uncertainty. Finally, we describe an active-learning-based labelling strategy to choose the unlabeled samples for manually labeling and retraining, thus reducing the cost of manually labeling a large quantity of unlabeled samples.

### 3.1. Overall Structure

The existing segmentation models based on deep learning generally require a large number of labeled images for training at substantial cost. Thus, it is crucial to achieve as high an accuracy as possible for segmentation with few labeled samples. To achieve this goal, we propose the overall framework as shown in Figure 1, which consists of three steps, i.e., the basic segmentation model, attention-based sampling and active-learning-based labelling.

During Step A, a labeled set of breast cancer images are first used to train the basic segmentation model, which obtains distinguished feature maps via multi-scale feature fusion and enhancement. In Step B, uncertainty sampling is first adopted to classify good and bad segmentation results for breast cancer in mammograms. Then, a novel attention mechanism is built to calculate weights for samples of the unlabeled breast cancer set. During Step C, unlabeled samples with higher weights, implying that they are more informative for learning, are selected to be manually labeled by professional medical experts. All these samples can then be further used to retrain the breast cancer segmentation model, thus boosting performance in an iterative manner.

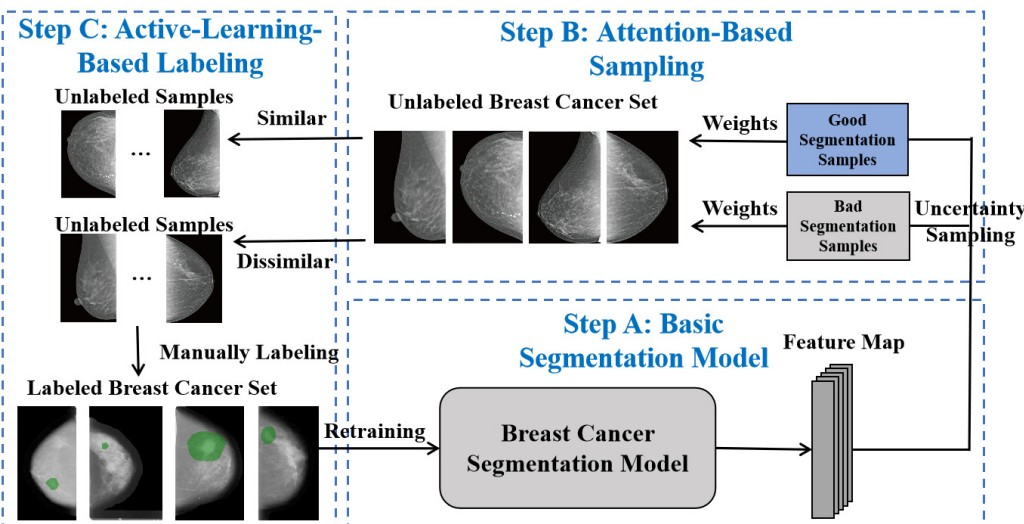

**Figure 1.** Framework of the proposed framework, which consists of (**A**) Basic Segmentation Model, (**B**) Attention-based Sampling, and (**C**) Active-Learning-based Labeling.

On the basis of structure design, we propose a loss function to involve the traditional IoU loss $Loss_I$ and binary cross-entropy loss $Loss_B$, i.e., $Loss = Loss_I + Loss_B$. Specifically, IoU loss $Loss_I$ can be calculated via

$$Loss_I = 1 - \frac{\sum_{i \in I} p_i \widehat{p}_i}{\sum_{i \in I} p_i + \widehat{p}_i - p_i \widehat{p}_i}, \tag{1}$$

where $I$ refers to the input breast cancer image, $i$ represents each pixel in the image, and $\widehat{p}_i$ and $p_i$ represent the predicted and true labeled values for each pixel, respectively. Moreover, the binary cross-entropy loss $Loss_B$ can be calculated via

$$Loss_B = -\sum_{i \in I} p_i \log(\widehat{p}_i) + (1 - p_i) \log(1 - \widehat{p}_i), \tag{2}$$

Note that $Loss_I$ and $Loss_B$ are calculated in each iteration stage to achieve convergence of the training process.

### 3.2. Basic Segmentation Model for Breast Cancer

Although skip connections in UNet could avoid the loss of detailed information caused by continuous down-sampling, they cannot capture multi-scale information with strong restrictions on locality. To obtain multi-scale feature maps, we cascade the feature maps at multiple layers with different receptive fields, where skip connections are used across different layers.

On the basis of the UNet segmentation model, the proposed basic model further involves the strength of the promotion feature module (PFM) [32] for the features of each output layer, which fuses multi-scale feature maps to enhance their representation capability. Note that the PFM works as a feature fusion and enhanced block in our former work, which not only fuses the features from multiple scales, but also selectively forgets useless information and enhances informative information, thus constructing more effective feature representation.

Specifically, an input image $I$ in the labeled breast cancer set is sent to the proposed basic segmentation model for feature extraction:

$$F^i = Seg_t^i(I), \tag{3}$$

where $F_i$ refers to feature maps corresponding to the $i$th output layer of the UNet model, and the function $Seg^t()$ refers to the segmentation model during the $t$th iteration of retraining.

Then, feature maps of multiple layers are sent to different PFMs for fusion and enhancement:

$$F_E^i = G_e^i(G_f^i(F^i)), where\ i = 1, 2, ..., n \tag{4}$$

where $F_E^i$ represents the output feature map after processing of the $i$th PFM, functions $G_e$ and $G_f$ represent enhance and forget operations in PFM, and $n$ is the total number of layers in the segmentation model.

In the later fusion step, the original feature map $F^i$ corresponding to the $i$-th layer is fused with its enhanced version $F_E^i$ via skip connections:

$$F^{i-1} = F_E^i \oplus X_{UP}(F^i), where\ i = 1, 2, ..., n \tag{5}$$

where $F^{i-1}$ refers to the output feature map of the $i-1$th layer, and the function $X_{UP}()$ is an up-sampling operation.

Finally, the generated feature map $F_P^1$ combines the high-layer semantic features with shallow features, thus enhancing representation capability for segmentation via multi-scale feature extraction fusion. We set $n = 5$ for all experiments in this paper.

### 3.3. Attention-Based Sampling Scheme

To obtain more segmentation related knowledge with as few labeled samples as possible, it is essential to obtain more distinguished feature representation from the training stage. We thus judge the informativeness for one specific sample based on its segmentation result with the following equation:

$$Info_i = \begin{cases} 1, if\ U_i \geq \alpha \\ 0, otherwise \end{cases} \tag{6}$$

where $Info_i$ implies whether the $i$th sample is useful for learning knowledge or not, $\alpha$ is a pre-set parameter based on segmentation performance of experiments, and $U_i$ calculates similarity coefficient with the following equation:

$$U_i = \frac{S_{i,q} \cap S_{i,g}}{S_{i,q} \cup S_{i,g}}, \tag{7}$$

where $S_q$ and $S_g$ respectively represent the predicted and ground-truth feature maps of the $i$th breast cancer image, which is used to represent the uncertainty and guides the selection of unlabeled breast cancer images. We consider the segmentation result of the $i$-th breast cancer image as good only if $Info_i = 1$; otherwise, it is considered bad.

With such criteria for judging informativeness, we choose samples from an unlabeled breast cancer set, which are either dissimilar to good segmentation samples or similar to bad segmentation samples, thus greatly improving the learning capability of the segmentation model for the features of difficult samples. Essentially, a soft attention model is generally formed as a dimension of interpretability into internal representations by selectively focusing on specific information. The core procedure of soft attention model [33] can calculate weights based on similarity between an input signal and pre-trained weights. Therefore, we propose a novel attention mechanism which assigns weights based on similarity calculations between unlabeled and labeled samples. In other words, the proposed attention mechanism assigns smaller weights if the unlabeled samples are more similar to the good segmentation samples. On the contrary, it would give higher weights to unlabeled samples with greater similarity to bad samples.

Defining the input unlabeled breast cancer image as query $Q$ and the set of labeled segmentation samples as $W$, a multi-layer perceptron(MLP) is utilized to calculate the similarity or correlation between $Q$ and one of the pre-trained samples $W_i$ as $sim(Q, W_i) = MLP(Q, W_i)$.

Afterwards, we adopt the Softmax function to perform normalization on the calculated similarity and emphasize the informative parts based on their inherent ability:

$$\alpha_i = softmax(sim(Q, W_i)) = \frac{e^{sim(Q, W_i)}}{\sum_{j=1}^{L} e^{sim(Q, W_i)}}; \tag{8}$$

where $L$ refers to the number of samples in the labeled segmentation results.

### 3.4. Active Labeling Strategy

Essentially, we believe that unlabeled samples with larger attention weight could contribute to the classification capability of the segmentation model, thus boosting the segmentation performance of the model by using these samples for retraining. Therefore, we propose a labeling strategy for breast cancer samples based on an active learning method, as described in Algorithm 1.

Specifically, we sort the unlabeled samples by weights and choose the unlabeled samples with higher weights for manual labeling, represented as Step 8 in the algorithm. The specific processes of sample calculation, selection and manual labeling and retraining are shown in Figure 2. We use both similarity and dissimilarity weights to select unlabeled images. Then, these samples are first roughly labeled by automatic labeling software and then manually adjusted by experts. Afterwards, they are added as labeled samples into the set of labeled breast cancer images. Meanwhile, these samples are deleted from the unlabeled image set, which can be represented as

$$\begin{cases} U_{t+1} = U_t - I_t \\ L_{t+1} = L_t + I_t \end{cases} \tag{9}$$

where $t$ refers to the iteration time of training, and $L_t \cap U_t = \varnothing$ to ensure the consistent processing of different iterations.

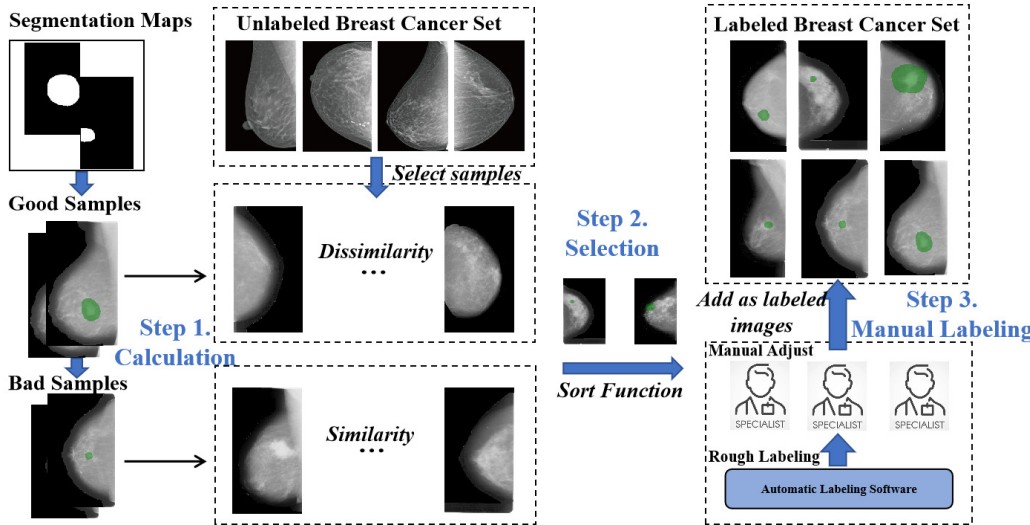

**Figure 2.** Key steps, i.e., Calculation, Selection, and Manual Labeling, in the proposed active-learning-based labeling strategy.

Finally, the newly constructed labeled set $L_{t+1}$ is used to retrain the basic segmentation model. With all these steps, an active learning iteration process is completed, where the performance of the breast cancer segmentation model could be improved step by step.

---

**Algorithm 1** Labeling strategy of breast cancer samples based on active learning.

---

**Require:** Unlabeled sample set $U$, labeled sample set $L$
**Ensure:** Labeled dataset $L_t$
　1: **While:**
　2: **if** $L \neq \varnothing$ **then**
　3:　　Train the breast cancer segmentation model
　4:　　Use the updated segmentation model to infer the labeled breast cancer images in the labeled sample set $L$, and output the feature maps $F_P$
　5:　　Calculate the similarity of the $i$-th image
　6:　　Calculate the attention weights
　7: **end if**
　8: Sort the breast cancer images in the unlabeled sample set according to attention weights

　9: Manually label the selected images which are assigned with higher weights
10: Update $U_{t+1}$ and $L_{t+1}$
11: Retrain the segmentation model for breast cancer images
12: **EndWhile**

---

## 4. Experimental Results

This section first introduces the datasets and measurements. Then, it describes the ablation experiments constructed to verify the effectiveness of the proposed modules. Afterwards, we demonstrate the performance of the segmentation models on four datasets to verify the effectiveness of the proposed framework. We also qualitatively compare the proposed active learning framework with existing methods. Finally, we offer the implementation details for readers' convenience.

### 4.1. Datasets and Measurements

We collected breast cancer images from a cooperating hospital, which are not released due to privacy reasons. The dataset includes 1462 labeled breast cancer images, with resolution $6781 \times 3676$ pixels . Since images were acquired by different scanners, we divided all our samples into four parts based on the type of scanner, i.e., Breast-A, Breast-B, Breast-C and Breast-D.

Various measurements were used to verify the effectiveness of the breast cancer image segmentation results, i.e., the mean Dice similarity coefficient (mDice), the mean intersection ratio (mIoU), and the mean absolute error (MAE). Where $TP$, $FP$ and $FN$ indicate true positive, false positive and false negative samples, mDice can be calculated as:

$$mDice = \frac{2TP}{FP + 2TP + FN}.$$ (10)

Note that a higher mDice implies a greater similarity between two samples.

IoU is defined as the area of the intersection divided by the area of the union of the predicted bounding box, which can be evaluated by

$$IoU = \frac{\text{area}\left(B_p \cap B_{gt}\right)}{\text{area}\left(B_p \cup B_{gt}\right)},$$ (11)

where $B_p$ and $B_{gt}$ are the predicted and ground-truth segmentation results, respectively.

By calculating the Euclidean distance between the predicted and the ground-truth results, MAE can be defined as:

$$MAE = \frac{1}{n} \sum_{i=1}^{n} |Y_i - \hat{Y}_i|,$$ (12)

where $n$ refers to the total number of samples, and $Y_i$ and $\hat{Y}_i$ are the predicted and ground-truth labels, respectively. Note that a lower MAE implies a better segmentation result.

### 4.2. Ablation Experiments

To evaluate the effectiveness of the attention-based sampling mechanism and the active labeling strategy, we designed several ablation experiments as shown in Table 1. Note that PFMs represent multiple promotion feature modules used in the basic segmentation model, Att refers to the proposed attention-based sampling scheme, and Act is the proposed active labeling strategy.

After adding PFMs, Att and Act, the segmentation performance on breast cancer images gradually improved on our four breast cancer image datasets, proving the effectiveness of these three modules. Specifically, we found that UNet+PFMs could achieve more precise boundaries of polyp regions and performed accurate segmentation when compared with the basic network (i.e., UNet). However, the shallow usage of boundary information without multi-scale refinement for boundary regions leads to uncompact performance towards larger and more regularized-shape polyp regions. In contrast, due to the usage of PFMs to extract a more distinguishing feature map by fusing multi-scale information, boundaries achieved by the proposed method were much more obvious with clear contour lines, thus providing better segmentation performance.

The ablation experiment on Att proved that the attention-based sampling scheme improved segmentation performance on all datasets. The attention-based design helped in effectively selecting more valuable samples for the further manual labeling process. It is beneficial to focus on the most informative unlabeled samples, which brings the feature information required by the current model for performance improvement with the fewest updating iterations.

Act, representing the active labeling strategy, enlarges size of the labeled breast cancer set, thereby improving the effect of tumor segmentation. Due to the guidance of optimized selection on unlabeled samples, we observed that informative samples for the current trained model were added to the labeled samples set for further retraining.

**Table 1.** Ablation experiments with different network structure designs on Breast-A, Breast-B, Breast-C and Breast-D datasets, where PFMs, Att and Act represent multiple promotion feature modules used in the basic segmentation model, the proposed attention-based sampling scheme and the proposed active labeling strategy, respectively.

| Dataset | Method | mDice | IoU | MAE |
|---|---|---|---|---|
| Breast-A | UNet | 0.356 | 0.266 | 0.018 |
| | UNet+PFMs | 0.412 | 0.332 | 0.013 |
| | UNet+PFMs+Att | 0.431 | 0.361 | 0.013 |
| | UNet+PFMs+Att+Act | **0.462** | **0.394** | **0.009** |
| Breast-B | UNet | 0.403 | 0.29 | 0.013 |
| | UNet+PFMs | 0.492 | 0.371 | 0.009 |
| | UNet+PFMs+Att | 0.515 | 0.382 | 0.112 |
| | UNet+PFMs+Att+Act | **0.543** | **0.401** | **0.006** |
| Breast-C | UNet | 0.553 | 0.429 | 0.033 |
| | UNet+PFMs | 0.677 | 0.512 | 0.026 |
| | UNet+PFMs+Att | 0.693 | 0.518 | 0.029 |
| | UNet+PFMs+Att+Act | **0.725** | **0.533** | **0.023** |
| Breast-D | UNet | 0.369 | 0.261 | 0.040 |
| | UNet+PFMs | 0.451 | 0.382 | 0.311 |
| | UNet+PFMs+Att | 0.478 | 0.396 | 0.335 |
| | UNet+PFMs+Att+Act | **0.512** | **0.422** | **0.027** |

### 4.3. Comparative Experiments

In this subsection, we describe our comparative experiments and present heatmap visualizations, segmentation results and the effectiveness of active learning.

Figure 3 shows the generated heatmap of breast cancer segmentation achieved by the proposed framework, the comparative method and the ground truth on four datasets. We used UNet in comparisons of heatmap visualization and segmentation results. The last layer of the network can generate a heatmap for each input breast cancer image, which can be used to generate segmentation results. Comparison of heatmaps shows that the proposed framework could more accurately identify the breast cancer region, and thus obtained a better performance in the segmentation task. Even in the case of a blurred image boundary, the heatmap implies that further segmentation results would maintain high accuracy by focusing on the dominant parts of cancer regions.

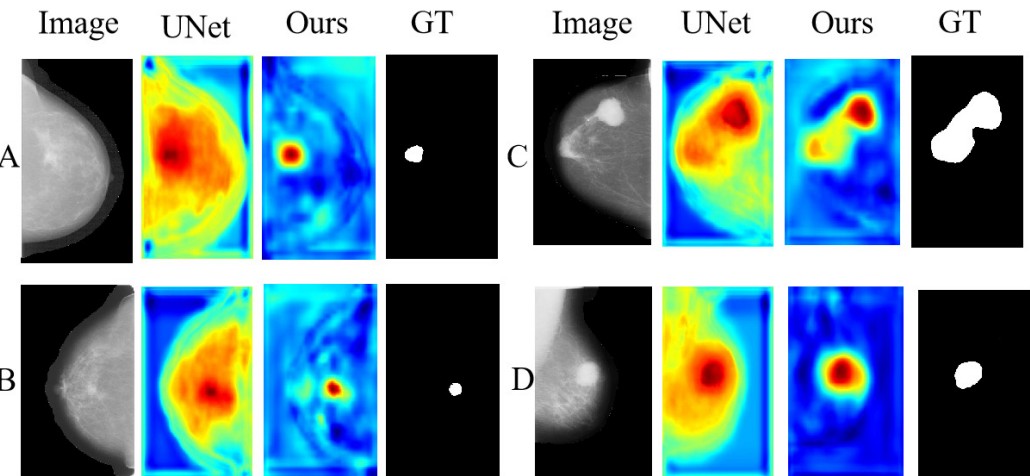

**Figure 3.** Heatmaps of breast cancer segmentation achieved by the proposed framework, the comparative method and the ground truth results. (**A**–**D**) refer to Breast-A, Breast-B, Breast-C and Breast-D datasets, respectively.

Figure 4 shows the qualitative comparison results of the breast cancer segmentation. Compared with the comparative segmentation model, i.e., UNet, the proposed framework achieved better segmentation results that were similar to the ground-truth results. Moreover, the proposed active learning strategy could refine the distinguished feature information of breast cancer using unlabeled samples, achieving more accurate pixel-level classification results. In addition, the proposed framework generates and refines the boundary region through an effective iterative update strategy, thus achieving global optimization progressively.

To verify that the active learning strategy can effectively reduce the cost of labeling, we conducted comparative experiments by selecting random sampling and CoreSet [34] as the comparison methods. Figure 5 shows plots of mean Dice for each iteration during the active learning. It is worth noting that the proposed active learning strategy not only had a higher mean Dice value, but also converged in fewer iterations. This proves that the proposed strategy selected more informative samples in each iteration, thus reducing the cost of labeling samples. Without measuring the uncertainty of sample labeling, other comparison methods might suffer from unstable convergence with increasing iterations because they adopt samples without helpful information, or even containing noisy information. Although all methods converged eventually, the compared methods tended to have lower mean Dice values due to the influence of noisy samples.

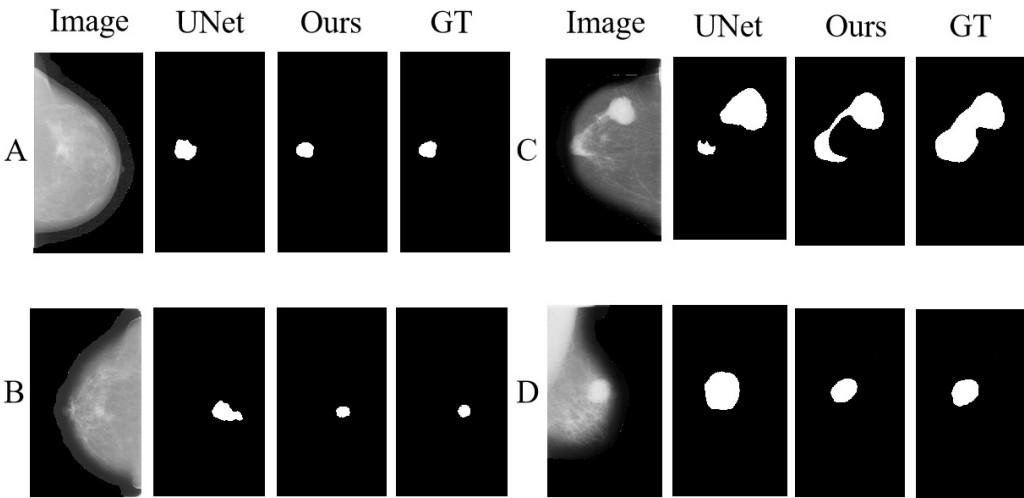

**Figure 4.** Qualitative comparisons between the segmentation results achieved by the proposed framework, the comparative method, and the ground truth. (**A**–**D**) refer to Breast-A, Breast-B, Breast-C and Breast-D datasets, respectively.

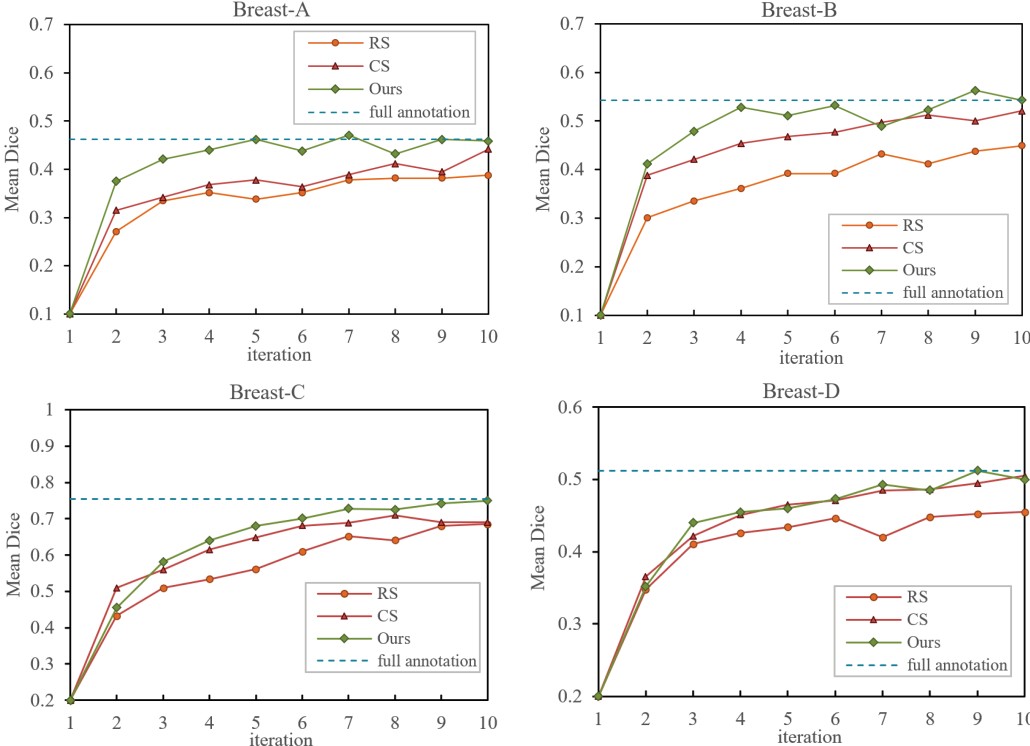

**Figure 5.** Plots of mean Dice coefficients, comparing different sampling strategies. It's noted that RS and CS refer to random selecting sampling and CoreSet sampling [34], respectively.

### 4.4. Implementation Details

Due to the scarcity of training samples, we used various data-enhancement methods to expand the training samples so that they met the requirements of model training. The size of input breast images was first adjusted to $352 \times 352$ pixels in the training and inference process. Then, we used image flipping to expand the number of training samples, in both horizontal and vertical directions. Finally, we not only randomly adjusted the contrast, brightness and sharpness of breast cancer images, but also randomly dilated and eroded image labels. All these operations were designed for data enhancement.

All experiments were carried out under the Linux Ubuntu operating system with a single Titan V GPU. We use Adaptive Moment Estimation (Adam) as the optimizer, while

the initial learning rate was set to 0.0001 and the learning rate was adjusted using learning rate decay. In the active learning strategy experiment , we used 100 unlabeled breast cancer images for initialization, and set the number of iterations to 10 for training of the basic segmentation model. Afterwards, we added the selected unlabeled samples to the labeled dataset, and trained 25 epochs for retraining in each iteration.

## 5. Conclusions

Due to the high cost of labeling training samples, herein we propose an attention-based active learning framework for the segmentation of breast cancer in mammograms. Specifically, we propose an attention sampling scheme to assign weights for unlabeled samples by evaluating their uncertainty. We also propose an active labeling strategy to select valuable unlabeled samples for manual labeling, thus enlarging the scale of the training set and improving the performance of the segmentation model. Testing on four datasets, experimental results showed that the proposed framework could greatly improve segmentation accuracy. The active learning scheme and attention strategy we adopted can be easily applied to other models and effectively reduce the data size required for model training.

In the future, we will try to introduce semi-supervised learning and unsupervised learning in active learning to further improve the generalization ability of the segmentation model on different datasets. Moreover, we will design specific algorithms to solve problems in breast cancer segmentation such as microcalcification and architectural distortion, thus improving the segmentation accuracy.

**Author Contributions:** Conceptualization, X.F. and Y.W. (Yirui Wu); methodology, X.F., H.H. and H.C.; software, X.F. and H.C.; validation, B.L., Y.W. (Yirui Wu) and Q.H.; formal analysis, X.F.; investigation, Y.W. (Yansong Wang); resources, X.F.; data curation, X.F. and Y.W. (Yirui Wu); writing—original draft preparation, X.F., H.C. and Y.W. (Yirui Wu); writing—review and editing, X.F., H.C. and Y.W. (Yirui Wu);visualization, B.L., Y.W. (Yansong Wang) and Q.H.; supervision, Y.W. (Yirui Wu) and Q.H.; project administration, Y.W. (Yirui Wu) and Q.H.; funding acquisition, X.F. and Y.W. (Yirui Wu). All authors have read and agreed to the published version of the manuscript.

**Funding:** This work was supported in part by a grant from General Scientific Research Project of Zhejiang Education Department (Y202147224), National Key R&D Program of China under Grant No. 2021YFB3900601, the Fundamental Research Funds for the Central Universities under Grant B220202074, the Fundamental Research Funds for the Central Universities, JLU, Joint Foundation of the Ministry of Education (No. 8091B022123).

**Informed Consent Statement:** Informed consent was obtained from all subjects involved in the study.

**Data Availability Statement:** Data available on request due to restrictions eg privacy or ethical.

**Conflicts of Interest:** The authors declare no conflict of interest.

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
