# Peer review of "Attention-Based Active Learning Framework for Segmentation of Breast Cancer in Mammograms"

_applsci, doi:10.3390/app13020852_

Round 1
Reviewer 1 Report
Title
The examination used for segmentation (mammography) is not reported in the title, but should be stated
The abstract is not clear about the imaging technique used for the study
Introduction
"Breast cancer has become the most serious malignant tumor that threatens the health of women in the world, while the incidence of breast cancer in our country increases gradually in recent years."
This sentence is too generic, not supported by numbers and references should be added
In the introduction I cannot see any reference, nor the background of the study, considering that many CAD systems are available for the automatic detection
I cannot get the role of Figure 1 in the introduction; moreover Fig 1 cleary demonstrates the inaccuracy of the segmentation.
The segmented areas are wider than the lesions, overestimating the lesions size.
In the figure on the superior right corner I suspect the presence of an area of architectural distortion, but this has not been considered in the segmentation.
The shown segmentations are coarse and imprecise
In Figure 5 I can see the segmentation of calcification, more than of effective lesion
Segmentation of breast lesion of mammography is very difficukt, as many lesions are visible just as microcalficiations or as architectural distortion.
The segmented examples shows the inaccuracy of the system, I have serious doubt on the reliability of the proposed segmentation system
Author Response
Response to Reviewer 1 Comments
Comment 1: The examination used for segmentation (mammography) is not reported in the title, but should be stated
Response 1: Thank you very much for your comment. We have replaced our title with " Attention-Based Active Learning Framework for Segmentation of Breast Cancer in Mammograms". As reviewer 1 stated, the images we used for segmentation are all mammograms, and all of our subsequent work is based on the mammograms already taken. In addition, we have realize that our earlier title was too general and did not reflect well the techniques used in the article. You can check the revised texts.
Comment 2: The abstract is not clear about the imaging technique used for the study
Response 2: Thank you very much for your comment. We have totally revised the abstract where now we explain the imaging technique used for study. You can check abstract for reading. Specifically, we first supplement the imaging technique used for the study, i.e., mammograms, and it is the most common way to screen for breast cancer. Then, we briefly introduce the basic segmentation network used in our paper, which is not mentioned in the previous version. Finally, We fixed some description errors in abstract and all these corrections can be found in the revised texts.
Comment 3: "Breast cancer has become the most serious malignant tumor that threatens the health of women in the world, while the incidence of breast cancer in our country increases gradually in recent years." This sentence is too generic, not supported by numbers and references should be added. In the introduction I cannot see any reference, nor the background of the study, considering that many CAD systems are available for the automatic detection.
Response 3: Thank you very much for your comment. We have added corresponding support data and references in the first sentence of Section 1, which proves the importance of conducting breast cancer research. In addition, we have realized that the lack of sufficient references in our introduction will make our paper unconvincing. Therefore, we added references where necessary. In fact, the background of our study is how to realize breast cancer segmentation without adequate annotation samples. In order to more clearly describe our research background, we deleted the sentence "how to…" in the first paragraph and pointed out the purpose of our research at the end of the third paragraph. Finally, our automatic detection framework is deployed in the few-shot scenario, i.e., using as few samples as possible for model training, where many existing CAD systems used for detection have poor performance. Therefore, it is necessary to study breast segmentation in this scenario.
Comment 4: I cannot get the role of Figure 1 in the introduction; moreover Fig 1 cleary demonstrates the inaccuracy of the segmentation. The segmented areas are wider than the lesions, overestimating the lesions size. In the figure on the superior right corner I suspect the presence of an area of architectural distortion, but this has not been considered in the segmentation.
Response 4: Thank you very much for your comment. In fact, the previous Figure 1 is intended to give an example of breast segmentation, both correct and incorrect segmentation. Therefore, in the previous figure 1, the reviewer could find that the segmented area was wider than the lesions area. During the common examination, the breast is photographed from different angles and several mammograms are generated to improve the accuracy of diagnosis. However, in our method, we mainly relied on the deep learning model to study lesions on mammograms, instead of considering the relationship between different mammograms. As a result, architectural distortions and other issues are not considered in our segmentation, and suspicious cases need to be examined further. To avoid ambiguity, we have removed the original Figure 1.
Comment 5: The shown segmentations are coarse and imprecise. In Figure 5 I can see the segmentation of calcification, more than of effective lesion
Response 5: Thank you very much for your comment. Due to our negligence, the previous segmentation could not well show the performance of our model, and we have re-selected the better segmentation results. All these corrections can be found in the Section 4.
Comment 6: Segmentation of breast lesion of mammography is very difficult, as many lesions are visible just as microcalficiations or as architectural distortion.
Response 6: I agree with the reviewer. Due to calcification and other problems, accurate segmentation of breast lesion of mammography is difficult. In fact, we often found that breast lesion and calcification were easily confused on lesion during our segmentation, which makes our segmentation accuracy still maintain at a low level. This problem is particularly acute in our model because each mammogram is treated as a separate sample. In future work, we plan to consider the relationship between different mammograms to mitigate the impact of this problem.
Comment 7: The segmented examples shows the inaccuracy of the system, I have serious doubt on the reliability of the proposed segmentation system.
Response 7: Thank you very much for your comment. We have selected segmentation results with higher accuracy. In theory, the strategies we adopted, i.e., the active labeling strategy and the attention sampling strategy, are proved to be feasible in other tasks. In the experiment, we believe that the proposed framework has achieved considerable segmentation effect under the scenario of scarce samples, and the previous segmentation samples cannot represent the level of the model. Admittedly, our model still has a lot of room for improvement and needs to be further improved.
Reviewer 2 Report
This paper proposes a novel breast cancer image segmentation based on deep active learning. The proposed method includes an attention sampling scheme and an active labeling scheme. The proposed active learning strategy in this paper can effectively reduce the labor of annotating labels, thus to improve the segmentation accuracy.
1) At the end of the abstract, it will be more intuitive and persuasive to verify its superiority and effectiveness through quantitative results of a large number of experiments.
2) The literature quoted in the introduction is old and cannot reflect the recent studies on active learning. Some recent studies need to be retrieved and cited to better reflect the innovation of this paper. And the comparison methods are relative old, please compare the proposed method with some newly published active learning based approaches.
3) The explanation of formula 6 is not sufficient and makes readers confused. Please further elaborate on the meaning of the formula.
4) In the absence of an appendix, Figure 2 represents the overview of the study, and enlarged Figure 2 can express the results more clearly and intuitively.
5)In the ablation studies, the influence of active learning round need to be further investigated.
6)Make sure your conclusions appropriately reflect on the strengths and weaknesses of your work, how others in the field can benefit from it, and thoroughly discuss future work.
Author Response
Response to Reviewer 2 Comments
Comment 1: At the end of the abstract, it will be more intuitive and persuasive to verify its superiority and effectiveness through quantitative results of a large number of experiments.
Response 1: Thank you very much for your comment. We have added the results to verify the superiority and effectiveness. Testing on four datasets, experimental results show that the proposed framework could greatly improve segmentation accuracy by about 15\% compared with the comparative study, while largely decreasing the cost of data annotation. Moreover, we also modified the abstract and correct wrong writings . You can check the revised texts.
Comment 2: The literature quoted in the introduction is old and cannot reflect the recent studies on active learning. Some recent studies need to be retrieved and cited to better reflect the innovation of this paper. And the comparison methods are relative old, please compare the proposed method with some newly published active learning based approaches.
Response 2: Thank you very much for your comment. We have updated the reference and discussed these works in the paragraph 4 of section 1. Different from other active learning methods, we use the attention mechanism to select informative samples, which further reduces the convergence speed of active learning in our paper. In addition, we add more reference in the introduction to make our paper convincing and persuasive. All these corrections can be found in the revised texts.
Comment 3: The explanation of formula 6 is not sufficient and makes readers confused. Please further elaborate on the meaning of the formula.
Response 3: Thank you very much for your comment. We have explained this formula with a reference, which explains the use of this formula and the setting of corresponding parameters. In fact, attention mechanism is adopted in our paper to compare the similarity and dissimilarity between unlabeled samples and segmented samples, for the guidance of subsequent sample selection. You can check the revised texts.
Comment 4: In the absence of an appendix, Figure 2 represents the overview of the study, and enlarged Figure 2 can express the results more clearly and intuitively.
Response 4: Thank you very much for your comment. We have enlarged Figure 2 and modified other figures. You can check the revised texts.
Comment 5: In the ablation studies, the influence of active learning round need to be further investigated.
Response 5: Thank you very much for your comment. We have added more discussions about the active learning in Section 4.4. In fact, active learning can help the model select more informative samples for model updating, which accelerates the speed of convergence. Therefore, we designed comparison experiments and selected unlabeled samples for model update during each iteration. Our experimental results in Figure 5 prove that sample selection with definite purpose (active learning) is more effective in improving accuracy. More analysis have been added in Section 4.4. You can check the revised texts.
Comment 6: Make sure your conclusions appropriately reflect on the strengths and weaknesses of your work, how others in the field can benefit from it, and thoroughly discuss future work.
Response 6: Thank you very much for your comment. We have offered more descriptions on the strengths and weaknesses of our work in Section 5. In fact, active learning and attention mechanisms are common strategies used in various computer vision tasks, where other methods can benefited for specific tasks with simple modifications. In addition, we added more discussions about the future work in the conclusions. All these corrections can be found in the revised texts.
Round 2
Reviewer 1 Report
The authors responded to my comments
Reviewer 2 Report
This manuscript can be published at its current version.